# Targeted Metabolomics Analysis Suggests That Tacrolimus Alters Protection against Oxidative Stress

**DOI:** 10.3390/antiox12071412

**Published:** 2023-07-12

**Authors:** Marie Joncquel, Julie Labasque, Julie Demaret, Marie-Adélaïde Bout, Aghilès Hamroun, Benjamin Hennart, Mathieu Tronchon, Magali Defevre, Isabelle Kim, Alain Kerckhove, Laurence George, Mylène Gilleron, Anne-Frédérique Dessein, Farid Zerimech, Guillaume Grzych

**Affiliations:** 1CHU Lille, Centre de Biologie Pathologie Génétique, Service Hormonologie Métabolisme Nutrition Oncologie, F-59000 Lille, France; 2CHU Lille, Centre de Biologie Pathologie Génétique, Institut d’Immunologie, F-59000 Lille, France; 3UMR1167 RIDAGE, Institut Pasteur de Lille, Inserm, Université de Lille, CHU Lille, F-59000 Lille, France; 4CHU Lille, Centre de Biologie Pathologie Génétique, Service Toxicologie et Génopathies, F-59000 Lille, France; 5Institut Pasteur de Lille, Université de Lille, ULR 4483, IMPECS, F-59000 Lille, France

**Keywords:** pipecolic acid, immunosuppressive, tacrolimus, metabolomics, oxidative stress

## Abstract

Tacrolimus (FK506) is an immunosuppressant that is experiencing a continuous rise in usage worldwide. The related side effects are known to be globally dose-dependent. Despite numerous studies on FK506, the mechanisms underlying FK506 toxicity are still not well understood. It is therefore essential to explore the toxicity mediated by FK506. To accomplish this, we conducted a targeted metabolomic analysis using LC−MS on the plasma samples of patients undergoing FK506 treatment. The aim was to identify any associated altered metabolic pathway. Another anti-calcineurin immunosuppressive therapy, ciclosporin (CSA), was also studied. Increased plasma concentrations of pipecolic acid (PA) and sarcosine, along with a decrease in the glycine/sarcosine ratio and a tendency of increased plasma lysine was observed in patients under FK506 compared to control samples. Patients under CSA do not show an increase in plasma PA compared to the control samples, which does not support a metabolic link between the calcineurin and PA. The metabolomics changes observed in patients under FK506 highlight a possible link between FK506 and the action of an enzyme involved in both PA and sarcosine catabolism and oxidative pathway, the Peroxisomal sarcosine oxidase (PIPOX). Moreover, PA could be investigated as a potential biomarker of early nephrotoxicity in the follow-up of patients under FK506.

## 1. Introduction

The use of immunosuppressants (IS) is constantly increasing in the world, with extensive indications: preparation and follow-up of transplants and immune system pathologies. For example, a recent US national cohort study of insured adults showed that more than 80,000 patients (2.8%) experienced drug-induced immunosuppression during the study period (2017–2019) [1]. There are several pharmacological classes, and among the immunosuppressants calcineurin modulator, Tacrolimus (FK506) are among the most used [1].

FK506 is a calcineurin inhibitor that forms a complex with FKBP12 on the calcium-calcineurin-NFAT (CCN) pathway [2]. This inhibition of calcineurin phosphatase activity leads to a decrease of T cell proliferation and pro-inflammatory signals. The CCN pathway is also inhibited by another IS: ciclosporin (CSA). Ciclosporin has the same mechanism of action as FK506 but forms a different complex with another immunophilin: cyclophilin. FK506 has several indications and is widely used [1,2]. However, its efficacy window is quite narrow, and maintaining target blood FK506 concentration remains difficult. Even when FK506 concentrations are within the therapeutic range in clinical settings, some patients experience rejection or toxicity. As a result, anorexia and organ toxicity are common. The most serious side effects of FK506 are nephrotoxicity, neurotoxicity, diabetogenesis, gastrointestinal disturbances, and hypertension. These side effects are globally dose-dependent. Unfortunately, despite numerous studies on FK506 and hypotheses on its mechanisms, including metabolite-related toxicity, accelerated apoptosis, and increased inflammation [3,4,5,6], the mechanisms related to its toxicity remain misunderstood. Hence, it is essential to explore FK506 mediated toxicity.

Among potential mechanisms, metabolic alteration could impact biological functions related to FK506 toxicity such as inflammation and energy imbalance [7,8]. Indeed, for instance, tacrolimus alters mitochondrial function in cultured human umbilical vein endothelial cells (HUVEC). Part of tacrolimus toxicity and vascular dysfunction may derive from metabolic alterations. The literature suggests that such effects alter energy metabolism in various tissues with high oxidative demand, and may be linked to increased oxidative stress [9]. Deciphering the metabolic alteration related to FK506 could lead to new information to determine the pathogenesis and FK506-mediated toxicity axis [10]. Xiao and colleagues compared the urinary metabolic profile of healthy volunteers and kidney transplant patients with tacrolimus-induced nephrotoxicity [7], and demonstrated that dimethylarginine and symmetric serine were biomarkers of renal injury. Furthermore, symmetric dimethylarginine showed a close correlation with serum creatinine, indicating renal function. However, this study was conducted solely on urine and did not reveal the changes in the patients’ blood, which is more representative of intermediate cellular metabolism. One study conducted by Zhu and colleagues [8] analyzed a set of plasma samples from patients with a liver transplant receiving Tacrolimus. The study revealed that dose-adjusted tacrolimus trough concentration was associated with 31 endogenous metabolites, including acylcarnitines, microbiota-derived uremic retention solutes, bile acids, and steroid hormones. However, as highlighted by the authors, the findings of untargeted metabolomics could not be confirmed by targeted analyses using internal standards. A recent multi-omics study on porcine proximal tubule cells exposed to tacrolimus [11] also showed deregulation of the oxidative stress balance by modification of the pathways involving amino acids. Changes in internal metabolites also suggest modifications in the Krebs cycle and in intermediary metabolism. However, this study suggests an impact of tacrolimus on the urea cycle and gluthation pathways. Unfortunately, the absolute quantification of metabolites in this study does not reveal the full range of differences between the conditions. However, this indicates that it is interesting and necessary to investigate tacrolimus-induced metabolic changes in order to predict these consequences. Therefore, it is important to employ a rigorous quantitative method to ensure the robustness of the results. Herein, our aim was to elucidate the metabolic alteration of FK506 intake using the robust metabolomic method.

To achieve this, we performed a quantitative LC−MS targeted metabolomic analysis using the internal standard on plasma of patients with FK506 treatment, to identify the altered metabolic pathway that could be responsible for the potentially adverse effects. Secondly, patients on ciclosporin treatment were also studied, as ciclosporin is also a calcineurin inhibitor but binds to a different complex than FK506. This second part of the study of patients under ciclosporin will allow us to better understand the links between the metabolic pathways and mechanisms of action of immunosuppressive molecules.

## 2. Materials and Methods

### 2.1. Description of the Patients

Patients with IS treatment (FK506 or CSA) and who were treated in the Lille University Hospital were retrospectively included in the study. The patients selected had no metabolic pathology (peroxisomal disease, antiquitin deficiency) or organ failure (kidney or hepatic), which could lead to alteration of the plasma amino acids. The IS group was compared with a control group comprising patients with the same inclusion criteria as the IS group, but who had not received any IS treatment. In line with the regulations set out by the French National Data Protection Commission and international recommendations [12], written informed consent was not required for this non-interventional retrospective study. There is only a retrospective data extraction in this manuscript, with no patient intervention, as the authors relied on the hospital database to extract the necessary biological data for the study. All patients had biological examinations, notably measurements of blood levels of IS and amino acids, as part of their routine care. Therefore, no additional tubes were collected for this study.

### 2.2. Plasma Amino Acid Measurements

Blood samples were collected in ethylenediamine tetra-acetate–coated tubes. Targeted metabolomics was performed to quantify plasma amino acids using the aTRAQ™ Kit (Sciex, Framingham, MA, USA), as described previously [13]. Briefly, plasma amino acids were amine-modified using the aTRAQ™ labeling reagent, which provides a specific mass tag for tandem mass spectrometry (MS). The tag is identified by MS/MS fragmentation of samples and standards while in multiple reaction monitoring mode. An internal standard (IS) set of aTRAQ™ labeled amino acids was used for detection and quantification by high performance liquid chromatography (Shimadzu UFLC-XR, C18 column, Kyoto, Japan) associated with MS/MS (Sciex 3200 Qtrap, Framingham, MA, USA) [13,14]. Quantification was performed by dividing the peak area for each analyte by the peak area of the corresponding IS and multiplying by the IS concentration.

### 2.3. Pipecolic Acid Measurements

Pipecolic acid was measured by HPLC/MS/MS. Briefly, a 6-point calibration range of PA (0–100 nmol/mL) was performed in serum albumin to avoid a possible matrix effect. A total of 40 µL of sample was mixed with 300 µL of a solution containing 1.4 mL 0.5 mM deuterated AP (D9) in 100 mL methanol to precipitate the proteins. The mixture was centrifuged twice for 10 min at 10,000× *g*. One hundred µL of the supernatant obtained was evaporated to dryness under nitrogen at 37 °C. Butylation was performed by adding 250 µL of butanol/HCL 3N.

### 2.4. Immunosuppressant Measurements

Immunosuppressants were quantified using a microparticle chemiluminescence immunoassay technique [15]. Briefly, a pre-analytical step was performed: 200 µL of whole blood on EDTA was mixed with 200 µL of a precipitation solution containing zinc sulfate in methanol and ethylene glycol. The mixture was centrifuged for 5 min at 13,000× *g*. The supernatant was then transferred to a pre-treatment tube before being placed in the ArchitectTM automated system. A chemiluminescent reaction was then observed and expressed in relative light units (RLU), with an indirect relationship between the quantity of IS and the RLU detected. The reference ranges were 5 to 15 ng/mL for blood FK506 and 100 to 300 ng/mL for blood CSA.

### 2.5. Statistical Analysis

Values are expressed as mean  ±  standard deviation (SD), mean  ±  standard error of the mean (SEM), or median  ±  inter-quartile range (IQR), as indicated in the figure legends. Statistical differences in clinical and biological parameters were assessed using the Mann–Whitney U or Student t test for continuous variables or χ2 test for categorical values. The Spearman rank test was used to determine the correlation between the variables and ordinal groups. Values of *p* < 0.05 were considered statistically significant. Principal component analysis was achieved with the R package FactoMineR. Random forest analyses were used to obtain a variable importance plot for discrimination of treatment status for each plasma amino acids species [16].

## 3. Results

### 3.1. Patients Characteristics

The selection of patients according to the previously established criteria (n = 35) enabled the constitution of 3 groups. The control group comprised 7 patients (5 women and 2 men) with an average age of 36 ± 12 years. The group of patients undergoing FK506 treatment comprised 19 patients (7 women and 12 men) with an average age of 58 ± 20 years, and a plasma FK506 concentration of 18.65 ± 8.48 ng/mL. The group of patients undergoing CSA treatment comprised 9 patients (3 women and 6 men) with an average age of 45 ± 14 years, and a plasma CSA concentration of 681 ± 695 ng/mL (Table 1). There was a significant difference in patient age between the groups, hence, in order to limit this bias, we will consider age in statistical analyses as a potentially confounding variable.

### 3.2. Targeted Metabolomics Showed Specific Metabolic Signature of Patient under FK506

Plasma amino acid were quantified in the two groups of patients (Control vs. FK506). We performed a principal component analysis for amino acids showing a normal distribution of their values (Figure 1A,B). The two first major axes selected from the PCA (Dim1 and Dim2) explained up to 20.6% of the variations between the groups ranks and succeeded to visually separate the two groups, along a “South-South-East North-North-West” axis (Figure 1B). The loading plot (Figure 1B) showed the variation in each amino acid with respect to the spatial separation of the groups in the Dim1–Dim2 plane. We found that pipecolic acid, sarcosine, and aspartate, showed strong distinction along the group separation axis (Figure 1B). We then sought to identify the plasma AA that most efficiently discriminated the groups. Interestingly, the top 3 plasma AA identified by random forest analysis were aspartate, sarcosine, and serine) (Figure 1C). We next perform an enrichment metabolite analysis on the most separating amino acids determined by unbiased analysis and showed that Glycine, Serine, and Threonine pathway are the most representative to discriminate our groups (Figure 1D).

### 3.3. Plasma Pipecolic Acid Increase in Case of FK506 Treatment

Metabolomics analysis shows that the glycine, serine, and threonine metabolism pathway is a major contributor to the differences between the control group and patients on FK506 and in particular, Pipecolic Acid (PA). We then focused on PA concentrations, and found that plasma PA concentrations in FK506 patients were significantly higher than plasma PA concentrations in the control group (5.29 ± 5.09 nmol/mL vs. 1.38 ± 0.46 nmol/mL, *p*-value = 0.005, unpaired test) (Figure 2A and Table 1). To ensure that age was not a confounding factor in the increase in PA, a correlation analysis was performed, and no correlation was shown between age and plasma PA concentration (r = −0.04, *p*-value = 0.84). There is therefore a link between FK506 treatment and increased plasma PA concentration.

### 3.4. Inhibition of the Calcineurin Pathway Does Not Increase the Concentration of Pipecolic Acid

The potential metabolic impact of FK506 on PA could be linked to calcineurin inhibition. The impact on PA metabolism of another anticalcineurin IS, ciclosporin (CSA), was therefore investigated [17]. Plasma PA concentrations were determined by the quantitative method and compared between control and CSA groups. Plasma PA concentrations in the CSA group were not significantly different from plasma PA in control group 1 (1.97 ± 0.86 nmol/mL vs. 1.38 ± 0.46 nmol/mL, *p*-value = 0.18, unpaired test) (Figure 2B and Table 1). Plasma PA concentrations in the FK506 group were significantly higher than plasma PA concentrations in the CSA group (5.29 ± 5.09 nmol/mL vs. 1.97 ± 0.86 nmol/mL, *p*-value = 0.01, unpaired test) (Figure 2C and Table 1). To ensure that the graft type was not a confounding factor in the increase in PA, an ANOVA analysis was performed. No statistical relationship was shown between graft type and PA concentration (F = 1.77, *p*-value = 0.17). The in vivo presence of CSA therefore does not increase the plasma PA concentration, unlike FK506. Additionally, calcineurin inhibition does not appear to be responsible for any apparent change in PA metabolism.

### 3.5. FK506 Appears to Alter the Metabolic Pathways Involving PIPOX

The literature reports substrate similarity for a PA catabolizing enzyme, pipecolate oxidase (PIPOX), which is similar to sarcosine oxidase (SARDH) [18]. This enzyme is involved in both the catabolism of pipecolic acid and the catabolism of sarcosine to glycine [19]. In order to investigate the metabolic impact of FK506 on these pathways, the concentration ratios of plasma AAs were studied, with particular emphasis on lysine, sarcosine, and glycine. Plasma lysine concentrations in the FK506 group showed an increasing trend compared to the control (201 ± 50 nmol/mL vs. 167 ± 28 nmol/mL, *p*-value = 0.10, unpaired test) (Figure 3A and Table 1). This increasing trend (not significantly demonstrated here) may be explained by the fact that lysine is metabolically more distant from PIPOX than pipecolic acid (Figure 4), so the impact on its increase may be less important. Plasma sarcosine concentrations in the FK506 group were significantly higher than plasma sarcosine concentrations in the control group (4.5 ± 2.1 nmol/mL vs. 0.60 ± 0.2 nmol/mL, *p*-value < 0.001, unpaired test) (Figure 3B and Table 1). The presence of FK506 in vivo therefore increases plasma sarcosine concentration. The plasma glycine/sarcosine ratio of FK506 group is significantly lower than the control group (60.7 ± 25.7 vs. 467.7 ± 162, *p*-value < 0.001, unpaired test) (Figure 3D and Table 1). FK506 therefore appears to have an impact on metabolic pathways involving the PIPOX enzyme and may therefore be responsible for the observed plasma increases in PA (Figure 4).

## 4. Discussion

The aim of the study was to determine a metabolomic signature of FK506 intake in order to better understand its overall consequences. Our results show a specific metabolic profile, and in particular that the plasma concentration of pipecolic acid increases in patients taking FK506. The hypothesis of a link between calcineurin and PA metabolism has also been considered, but the lack of in vivo impact of another anticalcineurin, CSA, means that this cannot be confirmed.

Pipecolic Acid (PA) is an amino acids homologue of proline, found in biological fluids (plasma, urine, and cerebrospinal fluid). It is derived from the catabolism of lysine, which can either follow the saccharopine pathway, which is mitochondrial and dominant in the liver, or the PA pathway, which is peroxisomal and dominant in the brain. The increase in plasma PA concentration may be linked either to a defect in PA catabolism associated with peroxisomal disease, to antiquitin deficiency, or to increased catabolism of lysine to PA in hepatocellular failure (via a decrease in the hepatic saccharopine pathway). On the other hand, the metabolism of sarcosine and glycine in the mitochondria plays a vital role in various biochemical processes in humans. Glycine, a non-essential amino acid, can be synthesized from sarcosine through the action of the enzyme sarcosine dehydrogenase. Once formed, glycine can participate in several metabolic pathways within the mitochondria. One of the important metabolic pathways involving glycine is its conversion to serine, which is another essential amino acid. This conversion is catalyzed by the enzyme serine hydroxymethyltransferase and is crucial for the synthesis of important compounds such as nucleic acid bases and phospholipids. Serine can also be converted back to glycine through the action of the enzyme serine dehydratase, allowing for fine regulation between these two amino acids. Furthermore, glycine can be utilized in the Krebs cycle, also known as the citric acid cycle or tricarboxylic acid cycle. In this cycle, glycine can be converted to acetyl-CoA, which is then utilized in energy production in the form of ATP. This metabolic pathway is essential for optimal mitochondrial function and cellular energy production. Additionally, sarcosine also plays a significant role in mitochondrial metabolism. It can be converted to acetyl-CoA through a series of enzymatic reactions, thereby contributing to energy production within the mitochondria. Moreover, sarcosine can be metabolized to glycine by the enzyme sarcosine dehydrogenase, providing a means of regulation between these two amino acids.

The PA increase could potentially be the consequence of interference between one of FK506 metabolites and PA measurement methods. Indeed, FK506 has within its structure the structure of PA, which is also an essential component of its synthesis [18]. However, this increase is not linked to analytical interference, as demonstrated by the fact that in vitro addition of FK506 does not increase pipecolic acid concentrations. Moreover, the targeted metabolome analysis does not seem to support this interferent hypothesis. Indeed, our results show that plasma sarcosine concentrations increase significantly in patients taking FK506. The literature shows that sarcosine and PA share a common enzyme, pipecolate oxidase (PIPOX), also known as sarcosine oxidase [19]. An increase of plasma PA and sarcosine concentration may therefore be linked to a direct or indirect decrease in the activity of PIPOX by FK506, or to a modification of PIPOX gene expression by FK506. Moreover, the literature shows that PIPOX appears to play an important role in protection against oxidative stress [20], so a reduction in its expression could be the cause of some of the adverse effects of FK506. Moreover, the link between Tacrolimus and oxidative stress has also been highlighted by another study [21]. Jin and colleagues demonstrated in glioma cell models that treatment with Tacrolimus increased the concentrations of certain markers of oxidative stress, such as antioxidant status, hydrogen peroxide level, and malondialdehyde. These alterations in oxidative stress could be related to an impairment of PIPOX, leading to a deficiency against oxidative stress. Our data can therefore complement the multi omics work recently published on porcine proximal tubular cells [11], and propose a new pathway to explain the altered oxidative stress shown in the different models studied. The alteration of PIPOX expression could not be studied in the article by Aouad, where only an RNA analysis targeting two genes was performed, so measuring PIPOX activity could constitute an interesting new avenue of research in these preclinical models.

The use of preclinical models (murine or cellular models) may be considered in order to study ex vivo the possible metabolic action of FK506 on PIPOX, by measuring PA and AA of interest after the addition of FK506. On the other hand, the measurement of PIPOX enzyme activity or determination of transcriptional expression of the gene encoding this enzyme by transcriptomics in patients on FK506 would confirm the impact of FK506 on this enzyme.

One of the described side effects of FK506 is nephrotoxicity. The early markers of this adverse effect are currently being sought [7]. The link between metabolism and nephrotoxicity induced by tacrolimus is of significant interest in understanding the effects of this immunosuppressive drug on the kidneys. Alterations in tacrolimus metabolism can lead to increased drug exposure, thereby increasing the risk of renal damage. Furthermore, some tacrolimus-related metabolites could be identified as potential markers of nephrotoxicity. Understanding the molecular mechanisms involved in tacrolimus metabolism and their relationship with nephrotoxicity can help identify individual risk factors and tailor treatment strategies to minimize adverse effects on the kidneys in patients receiving this crucial medication after transplantation. PA could therefore be of interest as a potential biomarker of nephrotoxicity. However, given the small size of the cohort in this study, further prospective investigations are required.

Our study has certain limitations that we would like to highlight. Firstly, our sample size is limited due to the difficulty in recruiting patients who have both an amino acid profile and immunosuppressant dosage information. However, to enhance the robustness of our results, we have employed non-parametric statistical tests. Secondly, our study is purely observational and should be confirmed in preclinical models in order to conduct functional studies on the activity of PIPOX in the presence of Tacrolimus.

This study established a potential link between FK506 intake and increased plasma PA. Treatment with FK506 should therefore be considered in the biological interpretation of PA levels, in order to avoid misleading clinicians towards other pathologies such as metabolic or liver diseases.

## 5. Conclusions

Metabolomics changes observed in patients under FK506 highlight a possible link between FK506 and the action of an enzyme involved in both PA and sarcosine catabolism and oxidative pathway, the Peroxisomal sarcosine oxidase (PIPOX). This observational study on patients from hospital care will need to be confirmed by functional tests on pre-clinical models (cells or animal models) in order to be able to measure the activity of the PIPOX enzyme in the presence of Tacrolimus. Moreover, PA could be investigated as a potential biomarker of early nephrotoxicity in the follow-up of patients under FK506.

## Figures and Tables

**Figure 1 antioxidants-12-01412-f001:**
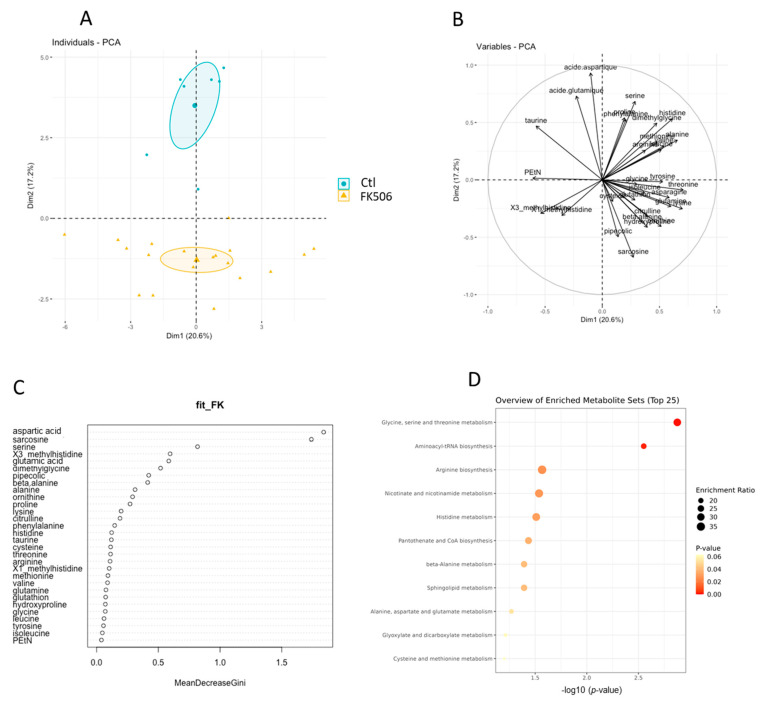
Unbiased analysis of plasma amino acid of patients with and without FK506. (**A**) Plot with Pearson correlation coefficients for each amino acid along the two major axis of the PCA; (**B**) Score plot of the groups along the two major axes of the PCA with 95% confidence ellipses; (**C**) Variable importance plot of the Random Forest analysis. The variables are ordered top-to-bottom as most-to-least important for classifying between the two groups; (**D**) Enrichment metabolite sets from separating more metabolites.

**Figure 2 antioxidants-12-01412-f002:**
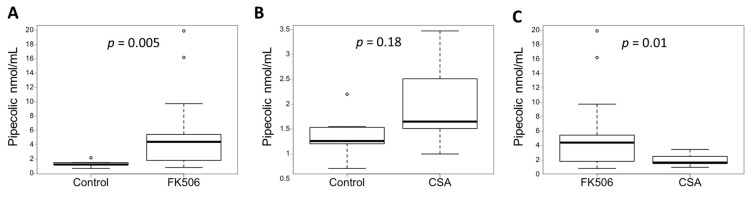
Plasma pipecolic acid increase in case of FK506 treatment independently of calcineurin pathway. (**A**) Plasma pipecolic concentrations according to different groups (FK506 vs Control). (**B**) Plasma pipecolic concentrations according to different groups (CSA vs Control). (**C**) Plasma pipecolic concentrations according to different groups (FK506 vs CSA) Control (n = 7), FK506 (n = 19) and CSA (n = 9). Data are expressed as median ± IQR. *p*-values are obtained using the Mann–Whitney test.

**Figure 3 antioxidants-12-01412-f003:**
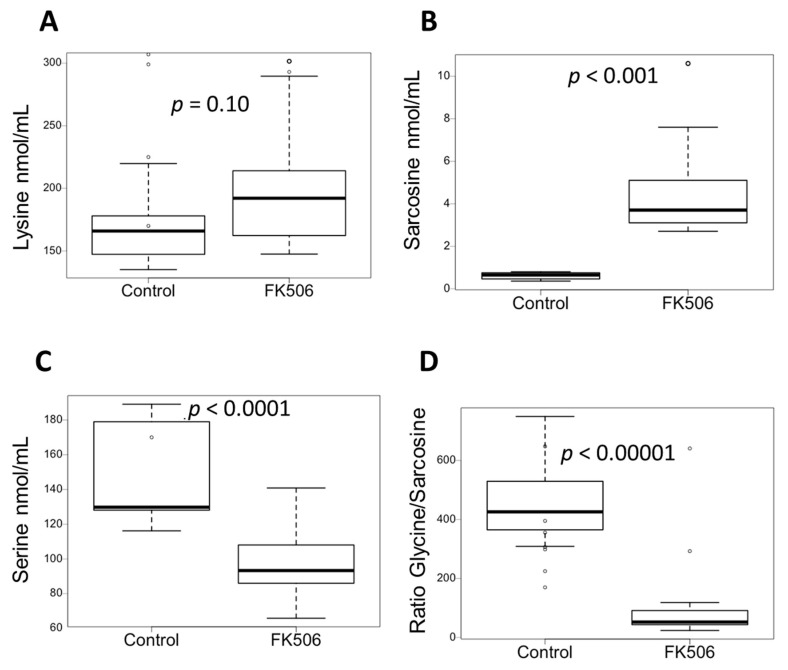
FK506 seems to alter the metabolic pathways involving PIPOX. (**A**) Plasma Lysine concentrations according to the different groups: Control (n = 7) and FK506 (n = 19). (**B**) Plasma Sarcosine concentrations according to the different groups: Control (n = 7) and FK506 (n = 19). (**C**) Plasma Serine, concentrations according to the different groups: Control (n = 7) and FK506 (n = 19). (**D**) Plasma Glycine/Sarcosine concentrations ratio according to the different groups: Control (n = 7) and FK506 (n = 19). Data are expressed as median ± IQR. *p*-values are obtained using the Mann–Whitney test.

**Figure 4 antioxidants-12-01412-f004:**
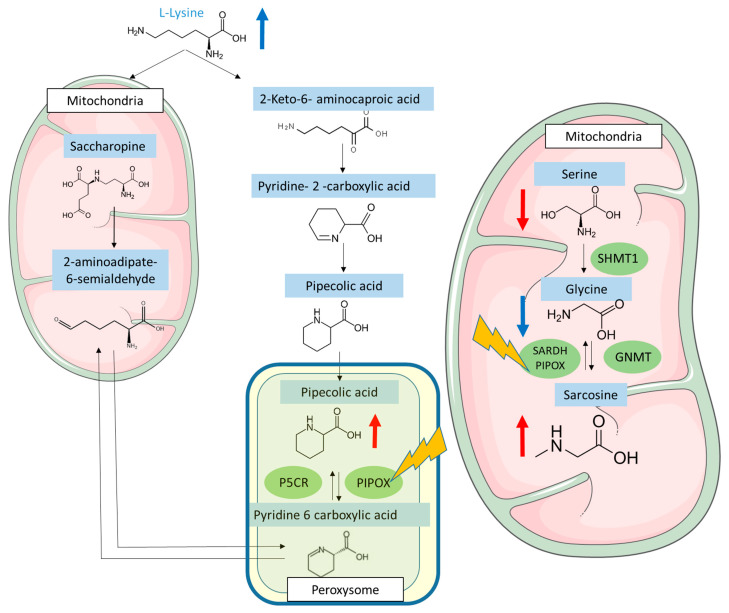
Summary of the metabolic changes induced by Tacrolimus. The red arrows indicate significant plasma variations, the blue arrows indicate non-significant plasma variation trends. PIPOX: pipecolate oxidase. SARDH: sarcosine oxidase. P5CR: pyrroline-5-carboxylate reductase. SHMT1: serine hydroxymethyltransferase. GNMT: glycine N-methyltransferase.

**Table 1 antioxidants-12-01412-t001:** Clinical and biological characteristics of selected patients. FK506: Tacrolimus, IS: immunosuppressants, PA: Pipecolic Acid, BM: Bone Marrow.

Patients	Control	FK506	CSA
n	7	19	9
Sex (Women/Men)	5/2	7/12	3/6
Age (years) Blood IS (ng/mL)	36 ± 12	58 ± 20	45+/14
	0	18.65 ± 8.48	681 ± 695
Type of graft (kidney/liver/heart/BM)	0/0/0/0	10/2/7/0	0/0/3/6
Plasma PA (nmol/mL)	1.38 ± 0.46	5.29 ± 5.09	1.97 ± 0.86
Plasma sarcosine (nmol/mL)	0.60 ± 0.17	4.5 ± 2.1	2.61 ± 2.58
Plasma lysine (nmol/mL) Plasma glycine (nmol/mL)	167 ± 28	201 ± 50	184 ± 28
	261 ± 45	234 ± 53	261 ± 75

## Data Availability

Data are available contacting corresponding author.

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
