# Peer review of "Targeted Metabolomics Analysis Suggests That Tacrolimus Alters Protection against Oxidative Stress"

_antioxidants, 2023, doi:10.3390/antiox12071412_

Round 1

Reviewer 1 Report

Minor revision: some errors found in the text

 Line 52: “FK506 has several indications and it widely used” the verb is missing

Line 102 Forty: why this number is written as a letter and the remaining as a number?

Mayor revision:

Results:

The number of patients is too small to be able to draw conclusions, the variability between patients cannot respond to robust statistical criteria. Factors such as age and sex are already different between groups, which can alter the results.

Why was the control group set up with so few patients? and, why does it not follow the same patient distribution as the treatment groups? It is curious that both the FK506 and CSA groups are about 64% male and the control group is 28% male.

Figure 2: why 3 graphs are presented instead of one with the three groups since the three groups are then compared with each other. In this way, everything could be plotted at the same scale and avoids visual confusion.

As mentioned in the discussion, the increase in PA may be due to the metabolism of tacrolimus, so its presence in the blood would simply confirm that the patient has taken this drug, I do not understand the association with nephrotoxicity shown in the conclusions. How is renal damage assessed? Do all patients develop the same kind of damage? The logical thing would be to make a correlation between PA and, for example, creatinine and/or GFR

Conclusions are drawn that are not experimentally proven.

The quality of the writing can be improved. A general revision of the text is recommended.

Author Response

Authors: We would like to thank the editor and the experts for the time they spent on our manuscript. We have taken all comments into account and hope that the manuscript will meet all the suggestions made to us.

Minor revision: some errors found in the text

Expert: Line 52: “FK506 has several indications and it widely used” the verb is missing

Authors responses: We would like to thank the expert for his vigilance and we changed our phrase.

Addition in the manuscript related to expert comment:

“FK506 is related to several indications and it widely used in medical field”

Expert:  Line 102 Forty: why this number is written as a letter and the remaining as a number?

Authors responses: We would like to thank the expert for his vigilance and we changed our phrase.

Addition in the manuscript related to expert comment:

40 µL of sample is mixed with 300 µL of a solution containing”

Mayor revision:

Results:

Expert: The number of patients is too small to be able to draw conclusions, the variability between patients cannot respond to robust statistical criteria. Factors such as age and sex are already different between groups, which can alter the results. Why was the control group set up with so few patients? and, why does it not follow the same patient distribution as the treatment groups? It is curious that both the FK506 and CSA groups are about 64% male and the control group is 28% male.

Authors responses: We thank the expert for this constructive comment. We are aware that one of the major limitations of the study is the small number of subjects we were able to include. However, there are a number of difficulties that may explain this low number: it is relatively rare for a plasma amino acid profile to be prescribed at the same time as an immunosuppressant assay. On the other hand, it is also very uncommon to perform a plasma amino acid assay in subjects without renal or hepatic metabolic pathologies (which were among the pathologies excluded from the control group in order to avoid any bias in plasma amino acid concentrations).

However, in order to obtain meaningful results, we used non-parametric statistical tests, which are more robust on small sample sizes: the minimum number of subjects per group for a non-parametric statistical test is 5. In addition, the p-values obtained are high, which means that we can be very confident in the results despite the small number of subjects.

We agree with the others that age and gender can constitute a confounding bias on certain results, so in this sense we have carried out additional statistical studies to be sure that they do not impact the result, as we have shown for example for the absence of a link between pipecolic acid and age in the manuscript.

With regard to unbiased non-targeted analyses (PCA and Random Forrest), the results presented in Figure 1 show a clear separation of groups in PCA. In view of the small number of patients, we combined this PCA analysis with a Random Forrest analysis to reinforce the significance of the results. Indeed, both unbiased analyses show a statistically validated metabolic pathway significance.

We have taken the reviewer's comments into account by adding them in a limitation paragraph.

Addition in the manuscript related to expert comment:

To ensure that age was not a confounding factor in the increase in PA, a correlation analysis was performed and no correlation was shown between age and plasma PA concentration (r = - 0.04, p-value = 0.84).”

“Our study, has certain limitations that we would like to highlight. Firstly, our sample size is limited due to the difficulty in recruiting patients who have both an amino acid profile and immunosuppressant dosage information. However, to enhance the robustness of our results, we have employed non-parametric statistical tests. Secondly, our study is purely observational and should be confirmed in preclinical models to conduct functional studies on the activity of PIPOX in the presence of Tacrolimus.”

Expert: Figure 2: why 3 graphs are presented instead of one with the three groups since the three groups are then compared with each other. In this way, everything could be plotted at the same scale and avoids visual confusion.

Authors responses: We thank the expert for this constructive comment. We initially hesitated to group the 3 graphs together, but separated them, as the concentrations of pipecolic acid move very little with CSA, and it was impossible to represent the median by grouping all the groups together. We also chose to separate the graphs to facilitate progressive reading in the manuscript, to help the reader understand the results presented.

Expert: As mentioned in the discussion, the increase in PA may be due to the metabolism of tacrolimus, so its presence in the blood would simply confirm that the patient has taken this drug, I do not understand the association with nephrotoxicity shown in the conclusions. How is renal damage assessed? Do all patients develop the same kind of damage? The logical thing would be to make a correlation between PA and, for example, creatinine and/or GFR

Authors responses: We thank the expert for this constructive comment. Our aim here is to provide perspectives for the practical application of our findings. Indeed, PIPOX could potentially be responsible for nephrotoxicity. Therefore, we suggest further investigations to propose potential early markers of nephrotoxicity.

Addition in the manuscript related to expert comment:

“One of the described side effects of FK506 is nephrotoxicity. Early markers of this adverse effect are currently being sought. The link between metabolism and nephrotoxicity induced by tacrolimus is of considerable interest in understanding the effects of this immunosuppressive drug on the kidneys. Alterations in tacrolimus metabolism can lead to increased drug exposure, thereby increasing the risk of renal damage. Furthermore, some tacrolimus related metabolites could be identified as potentially markers of nephrotoxicity. Understanding the molecular mechanisms involved in tacrolimus metabolism and their relationship with nephrotoxicity can help identify individual risk factors and tailor treatment strategies to minimize adverse effects on the kidneys in patients receiving this crucial medication after transplantation. PA could therefore be of interest as a potential biomarker of nephrotoxicity. However, given the small size of the cohort in this study, further prospective investigations are required.”

Expert: Conclusions are drawn that are not experimentally proven.

Authors responses: We thank the expert for this constructive comment. We have modified our introduction to show that pre-clinical functional studies are needed to confirm the results of this work

Addition in the manuscript related to expert comment:

This study established a potential link between FK506 intake and increased plasma PA. Treatment with FK506 should therefore be considered in the biological interpretation of PA levels, in order to avoid misleading clinicians towards other pathologies such as metabolic or liver diseases.

  1. Conclusions

Metabolomics changes observed in patients under FK506 highlight a possible link between FK506 and the action of an enzyme involved in both PA and sarcosine catabolism and oxidative pathway, the Peroxisomal sarcosine oxidase (PIPOX). This observational study on patients from hospital care will need to be confirmed by functional tests on pre-clinical models (cells or animal models) in order to be able to measure the activity of the PIPOX enzyme in the presence of Tacrolimus. Moreover, PA could be investigated as potential a biomarker of early nephrotoxicity in the follow-up of patients under FK506.

Reviewer 2 Report

The article presented by Marie Joncquel and collaborates, entitled “Targeted metabolomics analysis suggests that Tacrolimus alters 2 protection against oxidative stress”, is an original article that aimed to analyze the metabolomic toxicity of tacrolimus. The contribution of the manuscript to scientific literature is medium-low. It is a descriptive article and although the authors (15 authors) compare it with another immunosuppressant with similar characteristics, some functional study is missing.

Major revision:

1.       Line 57: “These side effects are globally dose dependent. 57 Unfortunately, despite numerous studies on FK506 and hypotheses on its mechanisms, 58 including metabolite-related toxicity, accelerated apoptosis, and increased inflammation 59 [3–6], mechanisms related to its toxicity remains misunderstood”. The introduction is very concise and does not speak of the state of the art of tacrolimus metabolomics. They only cite the sentence on line 57, which is very generic. A good article should review what is known up to now and reflect it in the text, it should not be referenced only.

2.       In the introduction the authors must introduce the metabolites analyzed: pipecolic acid (PA), sarcosine, gly-30 cine/sarcosine ratio, lysine..

3.       In the description of the patients is necessary to include additional information on concomitant medication, or are patients only taking tacrolimus or cyclosporine? If so, it should be reflected.

Author Response

Authors: We would like to thank the editor and the experts for the time they spent on our manuscript. We have taken all comments into account and hope that the manuscript will meet all the suggestions made to us.

Expert: The article presented by Marie Joncquel and collaborates, entitled “Targeted metabolomics analysis suggests that Tacrolimus alters 2 protection against oxidative stress”, is an original article that aimed to analyze the metabolomic toxicity of tacrolimus. The contribution of the manuscript to scientific literature is medium-low. It is a descriptive article and although the authors (15 authors) compare it with another immunosuppressant with similar characteristics, some functional study is missing.

Authors responses: We thank the author for his comment. The large number of authors is linked to the fact that many clinical and biological departments had to collaborate in view of the complexity of the problem: the department of metabolic biology, the department of nutrition, the department of immunology, the department of toxicology, the department of nephrology, we also had to call on statistical experts in view of the complexity of the multivariate analyses used in omics, the recruitment and selection of patients was a complex process. Our internal service policy means that all those who contributed to the work must be mention as co-authors of the paper. We want to preserve this recognition of everyone work, and not leave anyone out. This study is indeed observational and we agree that a functional study will be necessary to confirm this work. Unfortunately, we are located in a clinical hospital and do not have the tools to carry out these functional studies, which could then be carried out by expert techniques following publication of the manuscript. We have included this clarification in the manuscript in order to be able to confirm the expert's statements.

Addition in the manuscript related to expert comment: “This observational study on patients from hospital care will need to be confirmed by functional tests on pre-clinical models (cells or animal models) in order to be able to measure the activity of the PIPOX enzyme in the presence of Tacrolimus”.

Expert: Line 57: “These side effects are globally dose dependent. 57 Unfortunately, despite numerous studies on FK506 and hypotheses on its mechanisms, 58 including metabolite-related toxicity, accelerated apoptosis, and increased inflammation 59 [3–6], mechanisms related to its toxicity remains misunderstood”. The introduction is very concise and does not speak of the state of the art of tacrolimus metabolomics. They only cite the sentence on line 57, which is very generic. A good article should review what is known up to now and reflect it in the text, it should not be referenced only.

Authors responses: We thank the expert for this constructive comment, we have add more precision about these specific points.

Addition in the manuscript related to expert comment:

Among potential mechanisms, metabolic alteration could impact biological func-tions related to FK506 toxicity such as inflammation and energy imbalance [7,8]. Indeed, for instance, tacrolimus alters mitochondrial function in cultured human umbilical vein endothelial cells (HUVEC). Part of tacrolimus toxicity and vascular dysfunction may de-rive from metabolic alterations. The literature suggests that such effects alter energy me-tabolism in various tissues with high oxidative demand, and may be linked to increased oxidative stress [9]. Decipher metabolic alteration related to FK506 could provide new in-formation for the determination of pathogenesis and FK506-mediated toxicity axis [10]. Xiao and colleagues compared the urinary metabolic profile of healthy volunteers and kidney transplant patients with tacrolimus-induced nephrotoxicity [7], and demonstrated that dimethylarginine and symmetric serine were biomarkers of renal injury. Furthermore, symmetric dimethylarginine showed a close correlation with serum creatinine, indicating renal function. However, this study was conducted solely on urine and did not reveal the changes in the patients blood, which are more representative of intermediate cellular me-tabolism. One study on blood was conducted by Zhu and colleagues [8] on a set of plasma samples from patients with liver transplant receiving Tacrolimus and revealed that dose-adjusted tacrolimus trough concentration was associated with 31 endogenous me-tabolites, including acylcarnitines, microbiota-derived uremic retention solutes, bile acids and steroid hormones. However, as highlighted by the authors, the findings of untargeted metabolomics could not be confirmed by targeted analyses using internal standards. A very recent multi-omics study on porcine proximal tubule cells exposed to tacrolimus [11] also shows deregulation of the oxidative stress balance by modification of pathways in-volving amino acids. Changes in internal metabolites also suggest modifications in the Krebs cycle and in intermediary metabolism. However, this study suggests an impact of tacrolimus on urea cycle and gluthation pathways. Unfortunately, the absolute quantifi-cation of metabolites in this study does not reveal the full range of differences between conditions. However, this indicates that it is interesting and necessary to investigate tacro-limus-induced metabolic changes in order to predict these consequences. Therefore, it is important to employ a rigorous quantitative method to ensure the robustness of the re-sults. Herein, our aim was to elucidate the metabolic alteration of FK506 intake using robust metabolomic method.

Expert: In the introduction the authors must introduce the metabolites analyzed: pipecolic acid (PA), sarcosine, gly-30 cine/sarcosine ratio, lysine.

Authors responses: We thank the expert for this constructive comment. We agree that it is important to be able to help the reader by explaining the metabolism of the amino acids that emerge from the study. However, we prefer to include this explanatory section in the discussion rather than in the introduction. Indeed, the main aim is to carry out a non-targeted study, and these amino acids were only secondarily identified thanks to the Omics studies, so we have added these metabolite explanatory elements to the discussion in order to facilitate reading and understanding.

Addition in the manuscript related to expert comment:

Pipecolic Acid (PA) is an amino acids homologue of proline, found in biological fluids (plasma, urine and cerebrospinal fluid). It is derived from the catabolism of lysine, which can either follow the saccharopine pathway, which is mitochondrial and dominant in the liver, or the PA pathway, which is peroxisomal and dominant in the brain. The increase in plasma PA concentration may be linked either to a defect in PA catabolism associated with peroxisomal disease, to antiquitin deficiency, or to increased catabolism of lysine to PA in hepatocellular failure (via a decrease in the hepatic saccharopine pathway).

On the other hand, the metabolism of sarcosine and glycine in the mitochondria plays a vital role in various biochemical processes in humans. Glycine, a non-essential amino acid, can be synthesized from sarcosine through the action of the enzyme sarcosine dehydrogenase. Once formed, glycine can participate in several metabolic pathways within the mitochondria. One of the important metabolic pathways involving glycine is its conversion to serine, another essential amino acid. This conversion is catalyzed by the enzyme serine hydroxymethyltransferase and is crucial for the synthesis of important compounds such as nucleic acid bases and phospholipids. Serine can also be converted back to glycine through the action of the enzyme serine dehydratase, allowing for fine regulation between these two amino acids. Furthermore, glycine can be utilized in the Krebs cycle, also known as the citric acid cycle or tricarboxylic acid cycle. In this cycle, glycine can be converted to acetyl-CoA, which is then utilized in energy production in the form of ATP. This metabolic pathway is essential for optimal mitochondrial function and cellular energy production. Additionally, sarcosine also plays a significant role in mitochondrial metabolism. It can be converted to acetyl-CoA through a series of enzymatic reactions, thereby contributing to energy production within the mitochondria. Moreover, sarcosine can be metabolized to glycine by the enzyme sarcosine dehydrogenase, providing a means of regulation between these two amino acids.”

Expert:       In the description of the patients is necessary to include additional information on concomitant medication, or are patients only taking tacrolimus or cyclosporine? If so, it should be reflected.

Authors responses: We thank the expert for this constructive comment. We checked that patients were not taking any other immunosuppressive drugs or drugs likely to alter general metabolism (such as treatments for metabolic diseases (diabetes, dyslipidemia, etc.).

Round 2

Reviewer 1 Report

After the revision carried out, the authors have substantially improved the manuscript. Although the main limitation (number of samples) cannot be resolved, they have explained in detail why their results may be relevant.

Nothing to comment

Reviewer 2 Report

The authors have introduced the suggested changes